# Deep learning-based parameter mapping for joint relaxation and diffusion tensor MR Fingerprinting

**Carolin M. Pirkl**[*1,2]                                            CAROLIN.PIRKL@TUM.DE
**Pedro A. Gómez**[*1]                                               PEDRO.GOMEZ@TUM.DE
**Ilona Lipp**[3,4,5]                                                LIPPI@CARDIFF.AC.UK
**Guido Buonincontri**[6,7]                                          GUIDO.BUONINCONTRI@GMAIL.COM
**Miguel Molina-Romero**[1]                                          MIGUEL.MOLINA@TUM.DE
**Anjany Sekuboyina**[1,8]                                           ANJANY.SEKUBOYINA@TUM.DE
**Diana Waldmannstetter**[1]                                         DIANA.WALDMANNSTETTER@TUM.DE
**Jonathan Dannenberg**[2,9]                                         JONATHAN.DANNENBERG@TU-DORTMUND.DE
**Sebastian Endt**[1,2]                                              SEBASTIAN.ENDT@TUM.DE
**Alberto Merola**[3,5]                                              ALBERTO.MEROLA@AICURA-MEDICAL.COM
**Joseph R. Whittaker**[3,10]                                        WHITTAKERJ3@CARDIFF.AC.UK
**Valentina Tomassini**[3,4,11]                                      VALENTINA.TOMASSINI@UNICH.IT
**Michela Tosetti**[6,7]                                             MICHELA.TOSETTI@FSM.UNIPI.IT
**Derek K. Jones**[3,12]                                             JONESD27@CARDIFF.AC.UK
**Bjoern H. Menze**[†1,13,14]                                        BJOERN.MENZE@TUM.DE
**Marion I. Menzel**[†2,9]                                           MENZEL@GE.COM

[1] *Department of Informatics, Technical University of Munich, Garching, Germany*

[2] *GE Healthcare, Munich, Germany*

[3] *Cardiff University Brain Research Imaging Centre (CUBRIC), Cardiff University School of Psychology, Cardiff, United Kingdom*

[4] *Institute of Psychological Medicine and Clinical Neurosciences, Cardiff University School of Medicine, Cardiff, United Kingdom*

[5] *Max Planck Institute for Human Cognitive and Brain Sciences, Leipzig, Germany*

[6] *Fondazione Imago7, Pisa, Italy*

[7] *IRCCS Fondazione Stella Maris, Pisa, Italy*

[8] *Department of Neuroradiology, Klinikum rechts der Isar, Munich, Germany*

[9] *Department of Physics, Technical University of Munich, Garching, Germany*

[10] *Cardiff University School of Physics and Astronomy, Cardiff, United Kingdom*

[11] *Institute for Advanced Biomedical Technologies (ITAB), Department of Neurosciences, Imaging and Clinical Sciences, School of Medicine, University "G. d'Annunzio" of Chieti-Pescara, Chieti, Italy*

[12] *Mary McKillop Institute for Health Research, Faculty of Health Sciences, Australian Catholic University, Melbourne, Australia*

[13] *Center for Translational Cancer Research, Munich, Germany*

[14] *Munich School of BioEngineering, Garching, Germany*

---

[*] Contributed equally

[†] Contributed equally

## Abstract

Magnetic Resonance Fingerprinting (MRF) enables the simultaneous quantification of multiple properties of biological tissues. It relies on a pseudo-random acquisition and the matching of acquired signal evolutions to a precomputed dictionary. However, the dictionary is not scalable to higher-parametric spaces, limiting MRF to the simultaneous mapping of only a small number of parameters (proton density, T1 and T2 in general). Inspired by diffusion-weighted SSFP imaging, we present a proof-of-concept of a novel MRF sequence with embedded diffusion-encoding gradients along all three axes to efficiently encode orientational diffusion and T1 and T2 relaxation. We take advantage of a convolutional neural network (CNN) to reconstruct multiple quantitative maps from this single, highly undersampled acquisition. We bypass expensive dictionary matching by learning the implicit physical relationships between the spatiotemporal MRF data and the T1, T2 and diffusion tensor parameters. The predicted parameter maps and the derived scalar diffusion metrics agree well with state-of-the-art reference protocols. Orientational diffusion information is captured as seen from the estimated primary diffusion directions. In addition to this, the joint acquisition and reconstruction framework proves capable of preserving tissue abnormalities in multiple sclerosis lesions.

**Keywords:** Magnetic Resonance Fingerprinting, Convolutional Neural Network, Image Reconstruction, Diffusion Tensor, Multiple Sclerosis

## 1. Introduction

Magnetic Resonance Imaging (MRI) has emerged as a powerful diagnostic imaging technique as it is capable of non-invasively providing a multitude of complementary image contrasts. Commonly used routine MRI protocols however lack standardization and mainly present qualitative information. To infer comprehensive diagnostic information, image analysis therefore requires extensive postprocessing for co-registration, motion-correction etc., a problem that exponentiates in multi-contrast acquisitions. Hence, fully quantitative multi-parametric acquisitions have long been the goal of research in MR to overcome the subjective, qualitative image evaluation (Thust et al., 2018). Progressing from qualitative, contrast-weighted MRI to quantitative mapping, Magnetic Resonance Fingerprinting (MRF) has emerged as a promising framework for the simultaneous quantification of multiple tissue properties (Ma et al., 2013). It aims at inferring multiple quantitative maps – proton density, T1 and T2 relaxation times in general – from a single, highly accelerated acquisition. MRF is based on matching the signal time-courses, acquired with pseudo-random variation of imaging parameters, to a dictionary of precomputed signal evolutions. As the dictionary is typically simulated with fine granularity of all foreseeable parameter combinations, this places a substantial burden on computational resources (Weigel et al., 2010; Ganter, 2018). Due to these memory and processing demands, the use of a dictionary becomes infeasible in higher-parametric spaces like in case of diffusion tensor quantification.

Over the last years, first diffusion-weighted MRF techniques have been proposed (Jiang et al., 2016, 2017; Rieger et al., 2018). However, the transient nature of MRF signals makes them highly vulnerable to motion artifacts, especially when aiming at encoding the full diffusion tensor – a drawback long-known from diffusion-weighted SSFP (DW-SSFP) techniques (McNab and Miller, 2010; Bieri and Scheffler, 2013). Susceptibility to motion together with the exponential scaling of the dictionary size with the dimensionality of the parameter space pose a significant challenge for the computation of the diffusion tensor,

limiting diffusion-weighted MRF applications to the estimation of the mean diffusivity, captured by the apparent diffusion coefficient, so far.

Also, recent work on combining MRF acquisition schemes with deep learning-based approaches for parameter inference has demonstrated to outperform conventional template matching algorithms in terms reconstruction quality and computation time (Cohen et al., 2018; Golbabaee et al., 2019; Fang et al., 2019).

In this proof-of-concept-study, we combine a novel MRF-type sequence and a deep learning-based multi-parametric mapping to simultaneously quantify T1 and T2 relaxation, and orientational diffusion. This work presets three main contributions:

1. We first present an MRF scheme with embedded diffusion-encoding gradients along all three axes to encode orientational diffusion information, whilst simultaneously maintaining differential weightings to T1 and T2.

2. Inspired by the promising results of image quality transfer ideas (Tanno et al., 2017; Alexander et al., 2017), we take advantage of a convolutional neural network (CNN) to reliably reconstruct paramatric maps of T1, T2 and the full diffusion tensor from the acquired MRF image time-series. With standard diffusion tensor imaging (DTI) analysis, it is then possible to derive scalar diffusion measures, and to estimate the principal diffusion direction.

3. We evaluate our approach on healthy subjects and on a clinical cohort of multiple sclerosis (MS) patients with substantial modifications of the brain micro-structure.

## 2. Material and methods

### 2.1. Relaxation and diffusion-sensitized MRF sequence

Inspired by DW-SSFP-based techniques, we propose an MRF acquisition scheme (Figure 1) that is sensitized to relaxation and orientational diffusion: We extend the steady state precession MRF methodology (Jiang et al., 2015) and insert mono-polar diffusion-encoding gradients before each readout. To encode the full diffusion tensor, we sensitize the MRF signal to 30 diffusion directions as it evolves in the transient state. Along the acquisition train ($t = 1224$ repetitions, $32\,\mathrm{s/slice}$), we repeat each diffusion-encoding direction 34 times before applying the next diffusion gradient direction. Directions of the diffusion-encoding gradients are chosen based on the electrostatic repulsion algorithm (Jones et al., 1999) and have amplitudes $g_{x,y,z}$ with $-40\,\mathrm{mT/m} \leq g_{x,y,z} \leq 40\,\mathrm{mT/m}$ and a duration $\delta = 3\,\mathrm{ms}$ . Every 6 directions, we incorporate non-diffusion weighted, unbalanced gradients ($g_{x,y,z} = 1\,\mathrm{mT/m}$). In each repetition, we acquire an undersampled image with one arm of a variable density spiral. To sample the entire k-space, 34 spiral interleaves are required. The spiral arms are rotated with the golden angle from one repetition to the next. To increase sensitivity to diffusion, the spiral readout ($TE = 6\,\mathrm{ms}$) happens after the diffusion gradient, similar to DW-SSFP imaging (Buxton, 1993). We rely on an initial inversion pulse ($TI = 18\,\mathrm{ms}$) that is followed by a train of constant flip angles with $\alpha = 37°$. In the latter part of the sequence, repeating variable flip angle ramps ($0° \leq \alpha \leq 49°$) are applied. $TR$ is set constant during diffusion-encoding ($TR = 22\,\mathrm{ms}$) with longer waiting periods ($TR = 50\,\mathrm{ms}$) when changing diffusion-encoding directions. As the diffusion-sensitization accumulates over multiple

repetitions, each timepoint of the acquired image time-series has a unique combination of diffusion-weighting and T1 and T2 contrast.

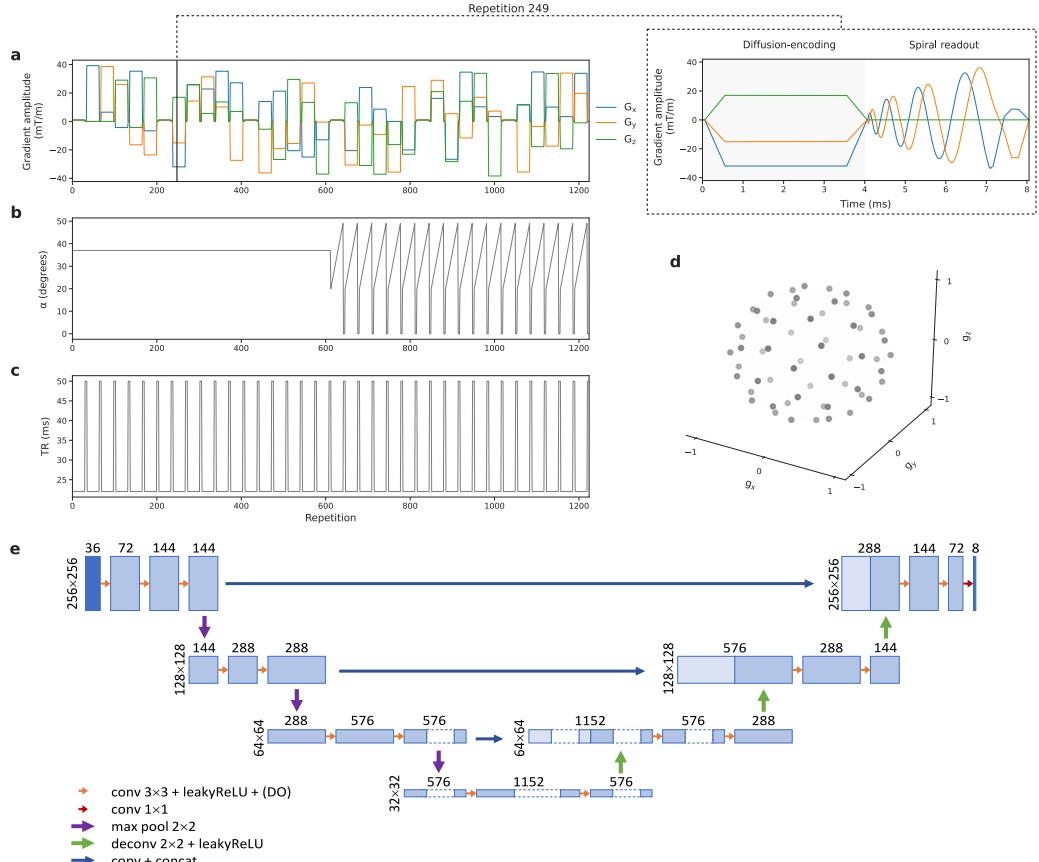

Figure 1: MRF acquisition and reconstruction framework. a) Pseudo-random variation of diffusion-encoding gradients. The diffusion-encoding direction is changed every 34 repetitions. In each repetition (here repetition 249), diffusion-encoding is followed by a spiral readout. b) Constant flip angles followed by variable flip angle ramps. c) TR pattern with longer waiting periods when changing diffusion-encoding directions. d) Spherical representation of the encoded diffusion directions. e) CNN architecture with the spatiotemporal magnitude MRF image series ($T = 36$ temporal channels) as input and quantitative maps of T1 and T2 relaxation times and the diffusion tensor elements ($Q = 8$ quantitative channels) as output.

## 2.2. Data acquisition and processing

As part of an IRB-approved study (Lipp et al., 2019), data from 11 MS patients and 9 healthy controls were acquired on a 3T HDx MRI system (GE Medical Systems, Milwaukee, WI) using an 8-channel receive-only head RF coil (GE Medical Devices), after ob-

taining written informed consent . The protocol included a single-shot EPI-DTI sequence, DESPOT1 (Deoni, 2007) and DESPOT2 (Deoni et al., 2003) sequences, high-resolution T2 and PD-weighted sequences, a FLAIR sequence for MS lesion segmentation, and a T1-weighted FSPGR sequence. In addition to these clinical sequences, 8-12 subsequent, axial slices, covering the middle portion of the brain, were acquired with the proposed MRF sequence. The main scan parameters of all acquisitions are shown in Table 2 in Appendix A.

**MRF image time-series**  We applied a sliding-window scheme (Cao et al., 2017) to reconstruct mixed-contrast images from consecutive spiral interleaves of the MRF acquisition. With a window size of 34, which corresponds to the spiral undersampling factor, we retrospectively fill up the undersampled k-space and reduce aliasing artifacts. We use a sliding-window stride of 34 to jointly reconstruct the consecutive images that were acquired with the same diffusion-encoding. By doing so, we reduce the dimensionality of the spatiotemporal MRF image data to $T = 36$ images along the temporal axis.

**Reference parameter maps**  Following the DESPOT1/2 approaches, we derived T1 maps from the SPGR and IR-SPGR images, and T2 maps from the phase-cycled bSSFP data. The EPI-DTI data were corrected for head motion, distortions induced by the diffusion-weighted gradients, and EPI-induced geometrical distortions by registering each diffusion image to the T1-weighted anatomical image using elastix (Klein et al., 2010). We then estimated the diffusion tensor with its diagonal $(D_{xx}, D_{yy}, D_{zz})$ and off-diagonal $(D_{xy}, D_{xz}, D_{yz})$ elements using ExploreDTI (Leemans et al., 2009). MS lesions were semi-manually segmented on the T2-weighted image, also consulting the FLAIR and the PD-weighted images using the Jim software package (Xinapse Systems). We obtained white matter (WM), gray matter (GM), and cerebrospinal fluid (CSF) masks from the lesion filled and brain-extracted T1-weighted images with FAST (Zhang et al., 2001). We then transformed the relaxation and diffusion tensor maps and the tissue segmentations to the MRF image space using ANTs (Avants et al., 2011), which incorporated a reorientation of the diffusion tensor images.

**Database**  With the processing pipeline described above, we created a database of 216 datasets in total, comprising both data from MS patients and healthy subjects. Each of the 216 datasets is a pair of the magnitude MRF image series $\mathbf{x} \in \mathbb{R}^{N \times N \times T}$ and the $Q = 8$ reference maps of T1, T2 and the 6 diffusion tensor elements $\mathbf{y} \in \mathbb{R}^{N \times N \times Q}$ with $N \times N = 256 \times 256$ being the spatial dimension.

## 2.3. CNN-based parameter mapping

We propose a CNN architecture to learn a non-linear relationship between the spatiotemporal MRF image data and multiple quantitative maps as an output. As such, the model presented in this work allows us to directly infer quantitative relaxation and diffusion information by capturing the temporal and neighborhood context features (Balsiger et al., 2019).

**CNN architecture**  For this multivariate regression, we propose a U-Net architecture (Ronneberger et al., 2015) which was previously shown to offer high quality parameter maps in MRF reconstruction (Fang et al., 2020) tasks. We implemented the convolutional-deconvolutional architecture as depicted in Figure 1 using TensorFlow. Our model receives

the spatiotemporal MRF magnitude image data $\mathbf{x}$ with its $T = 36$ temporal channels as input. In the contracting path, feature extraction is alternated by max-pooling to create a low-dimensional latent representation from the MRF image input. In the expansive path, the low-dimensional feature space is gradually decoded and upsampled to output quantitative maps $\mathbf{y}$ with $Q = 8$ parametric channels for T1, T2, and the 6 unique elements of the diffusion tensor. Using skip connections, the feature maps in the expansive path are concatenated with high-resolution feature maps from the contracting path, merging global context from the latent space with preserved spatial details from the input space.

**Data pre-processing** To foster effective network training, we normalized the magnitude MRF image series between $[0, 1]$ using its minimum and maximum intensities, $\mathbf{x}' = \frac{\mathbf{x} - \min(\mathbf{x})}{\max(\mathbf{x}) - \min(\mathbf{x})}$. To account for the widely varying scales for relaxation and diffusion tensor parameters, we transformed each quantitative map $\mathbf{y}_q$ to a fixed range of $\mathbf{y}'_q \in [0, 1]$ for T1, T2 and diagonal diffusion tensor maps, and $\mathbf{y}'_q \in [-1, 1]$ for off-diagonal diffusion tensor maps $\mathbf{y}'_q = \frac{\mathbf{y}_q}{\max(|q_{min}|, |q_{max}|)}$, using the global minimum and maximum parameter values $q_{min}$ and $q_{max}$. By doing so, we allowed directionality in the off-diagonal elements, captured as negative and positive value ranges, to equally impact the loss function. We also ensure that the loss function is implicitly balanced over all parameters and is not governed by the parameter with the highest magnitude, i.e. T1.

**Experimental setup** We trained the CNN for 400 epochs with a batch size of 5, using Adam optimization to minimize the L1 loss function with a learning rate of 0.0001, and a dropout rate of 0.25. For performance evaluation, we performed a 10-fold cross-validation on the 20 subjects, whereby each experimental instance consisted of 2 test subjects and 18 remaining subjects for training. Aiming at an efficient and robust reconstruction method, we increased the heterogeneity of the dataset as we ensured that training and testing datasets comprised both healthy subjects and MS patients. Network training for one instance of the cross-validation took 3.5 h on a Nvidia GeForce TITAN Xp GPU. The CNN training progress is illustrated in Figure 4 in Appendix B.

We applied standard DTI analysis to derive scalar diffusion metrics, i.e. mean diffusivity (MD), axial diffusivity (AD), radial diffusivity (RD), and fractional anisotropy (FA) from both the predicted and the reference diffusion tensors. To reflect the characteristic fiber orientation in WM, we obtained a colored FA map based on the primary diffusion eigenvector. We evaluated the reconstruction quality of our framework based on the structural similarity index measure (SSIM) and the root mean squared error (RMSE) between the CNN prediction and the DESPOT1/2 and EPI-DTI reference methods. To ensure comparability in terms of physical value ranges of the parameters, RMSE was derived from the normalized parameter maps $\mathbf{y}'$.

## 3. Results

It can be visually observed from Figure 2 that predicted relaxation and diffusion tensor maps are largely consistent with state-of-the-art methods, which is confirmed by the voxel-wise comparison in the difference maps. This is the case even though the input image series as obtained by the sliding-window reconstruction are impacted by artifacts due to

motion, undersampling and destructive interference between readout and diffusion-encoding gradients. Quantitatively, we achieved a comparable reconstruction performance for T1 and T2 with respect to DESPOT1/2 methods, while diagonal diffusion tensor elements show better agreement with the DTI reference than off-diagonal elements (Table 1). Specifically, it is more difficult for the CNN to reconstruct off-diagonal diffusion tensor information in WM and MS lesions than in GM and CSF. Overall, we reliably recovered diffusion and relaxation information, also in regions of diagnostic importance such as MS lesions, indicating generalization capability of our method. Figure 3 suggests that our framework is capable of reliably reconstructing diffusion information as the image quality of the scalar MD, AD, RD and FA maps is comparable to the EPI-DTI reference. The colored FA maps and the overlay of the primary eigenvectors of the predicted and reference diffusion tensors show that the principal diffusion direction and thus the characteristic fiber structure in WM is captured as illustrated by the enlarged portions of the derived maps. In both healthy WM tissue and MS lesions, RMSE suggests higher agreement with the reference maps for MD, AD and RD than FA (Table 1). This is in line with the overall SSIM which is higher for MD, AD and RD than for FA. Figure 5 in Appendix C depicts an exemplary dataset with significant artifacts due to patient motion. Here, the CNN is not able to successfully disentangle T1, T2 and diffusion information in severely corrupted regions.

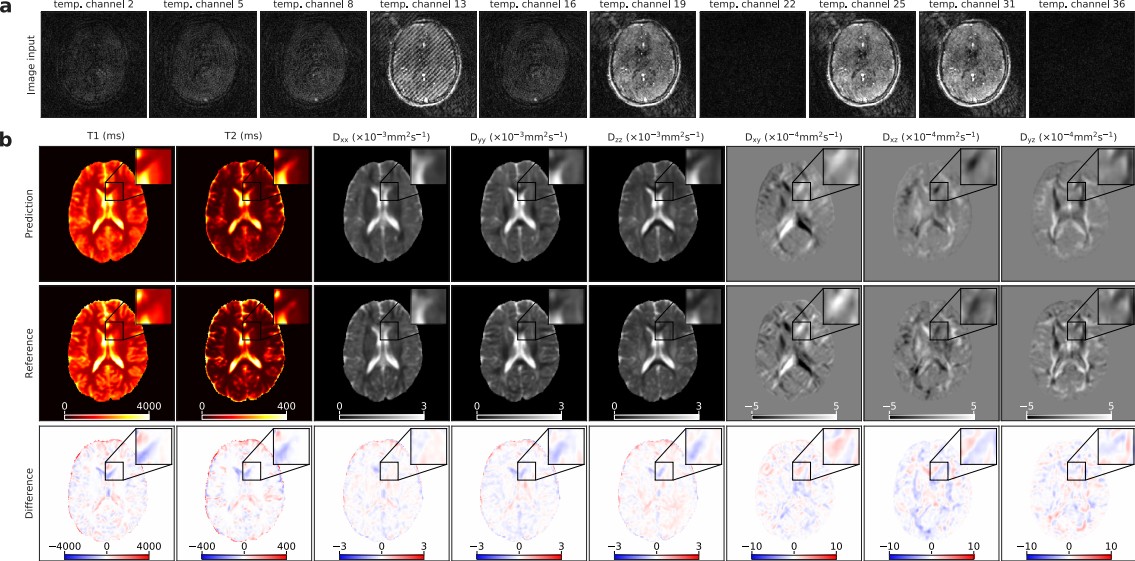

Figure 2: CNN reconstruction for a representative test dataset. a) Spatiotemporal MRF data show artifacts due to spatial undersampling, gradient interferences and motion. b) Predicted maps of T1, T2 and diffusion tensor elements do not show visual artifacts, providing satisfying image quality as demonstrated by the enlarged image sections. Voxel-wise difference maps do not reveal substantial differences between the proposed MRF framework and conventional reference methods.

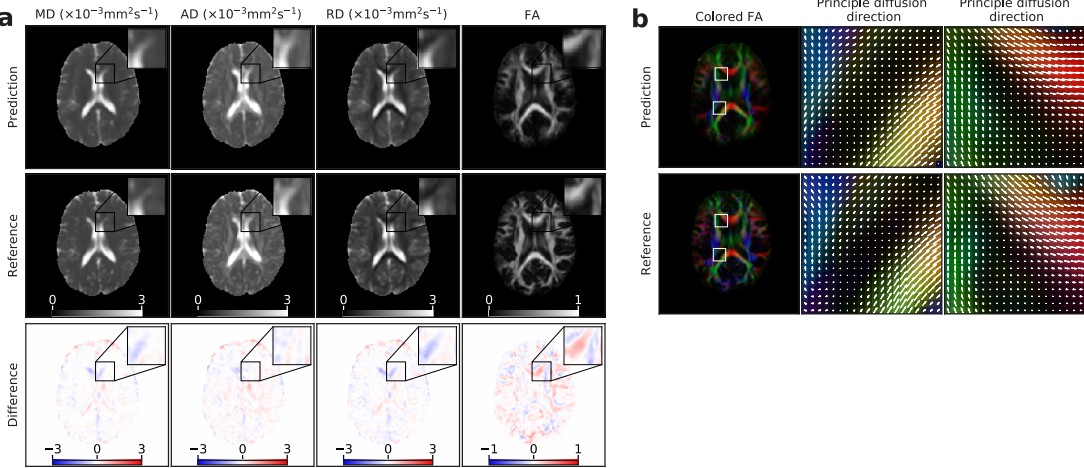

Figure 3: Diffusion tensor analysis for a representative test dataset. a) Scalar diffusion metrics, i.e. MD, AD, RD and FA, derived from predicted and reference diffusion tensors show good visual agreement. Voxel-wise difference maps and the enlarged views confirm that the reconstruction quality is largely consistent with the reference methods. b) Colored FA maps and the primary diffusion direction indicate the predominant diffusion direction in main WM-tracts.

Table 1: Quantitative comparison of our MRF framework with DESPOT1/2 and EPI-DTI reference methods.

| Metric | Region | Relaxation and diffusion tensor maps | | | | | | | |
|--------|--------|------|------|------|------|------|------|------|------|
| | | **T1** | **T2** | $\mathbf{D_{xx}}$ | $\mathbf{D_{yy}}$ | $\mathbf{D_{zz}}$ | $\mathbf{D_{xy}}$ | $\mathbf{D_{xz}}$ | $\mathbf{D_{yz}}$ |
| **SSIM** | — | $0.91 \pm 0.04$ | $0.9 \pm 0.04$ | $0.94 \pm 0.03$ | $0.94 \pm 0.03$ | $0.94 \pm 0.03$ | $0.83 \pm 0.05$ | $0.81 \pm 0.05$ | $0.82 \pm 0.05$ |
| **RMSE** | Whole brain | $0.15 \pm 0.06$ | $0.18 \pm 0.06$ | $0.07 \pm 0.03$ | $0.08 \pm 0.03$ | $0.07 \pm 0.03$ | $0.1 \pm 0.03$ | $0.11 \pm 0.02$ | $0.1 \pm 0.02$ |
| | CSF | $0.19 \pm 0.04$ | $0.25 \pm 0.06$ | $0.1 \pm 0.03$ | $0.1 \pm 0.03$ | $0.09 \pm 0.03$ | $0.07 \pm 0.02$ | $0.07 \pm 0.03$ | $0.07 \pm 0.02$ |
| | GM | $0.14 \pm 0.06$ | $0.16 \pm 0.07$ | $0.07 \pm 0.03$ | $0.07 \pm 0.03$ | $0.07 \pm 0.03$ | $0.08 \pm 0.02$ | $0.08 \pm 0.02$ | $0.08 \pm 0.02$ |
| | WM | $0.09 \pm 0.05$ | $0.09 \pm 0.07$ | $0.05 \pm 0.03$ | $0.05 \pm 0.03$ | $0.05 \pm 0.03$ | $0.12 \pm 0.04$ | $0.14 \pm 0.04$ | $0.13 \pm 0.04$ |
| | MS lesion | $0.11 \pm 0.07$ | $0.1 \pm 0.08$ | $0.06 \pm 0.04$ | $0.06 \pm 0.04$ | $0.06 \pm 0.04$ | $0.15 \pm 0.07$ | $0.17 \pm 0.06$ | $0.16 \pm 0.06$ |

| Metric | Region | Diffusion metric maps | | | |
|--------|--------|------|------|------|------|
| | | **MD** | **AD** | **RD** | **FA** |
| **SSIM** | — | $0.94 \pm 0.03$ | $0.93 \pm 0.03$ | $0.94 \pm 0.03$ | $0.91 \pm 0.03$ |
| **RMSE** | Whole brain | $0.11 \pm 0.05$ | $0.12 \pm 0.05$ | $0.11 \pm 0.05$ | $0.11 \pm 0.03$ |
| | CSF | $0.13 \pm 0.06$ | $0.14 \pm 0.06$ | $0.13 \pm 0.05$ | $0.08 \pm 0.06$ |
| | GM | $0.11 \pm 0.05$ | $0.11 \pm 0.06$ | $0.11 \pm 0.05$ | $0.09 \pm 0.03$ |
| | WM | $0.07 \pm 0.05$ | $0.08 \pm 0.04$ | $0.07 \pm 0.05$ | $0.13 \pm 0.03$ |
| | MS lesion | $0.09 \pm 0.07$ | $0.11 \pm 0.06$ | $0.1 \pm 0.07$ | $0.14 \pm 0.06$ |

## 4. Discussion and conclusion

In this work, we propose a relaxation and diffusion-sensitized MRF sequence combined with a CNN-based multivariate regression. We approach the underlying MRF sequence design and the deep-learning based parameter inference as a joint task. In this way, we can relax

the MR acquisition requirements and efficiently encode T1 and T2 relaxation times together with orientational diffusion information. With our joint MRF acquisition and reconstruction framework, we present a proof-of-concept of fast multi-parameter quantification. We simultaneously measure and reconstruct quantitative relaxation and diffusion tensor maps, significantly reduce the scan time (32 s/slice), and make extensive post-processing pipelines of conventional multi-contrast imaging redundant.

Following the concept of DW-SSFP imaging, the accumulation of T1 and T2-weighting and diffusion-sensitization along multiple repetitions and echo-pathways is what makes our signal encoding and hence the MRF acquisition so efficient. However, this is also what challenges the reconstruction the most: First, the signal dependence on T1, T2 and flip angles is relatively complicated. This impedes diffusion quantification in DW-SSFP approaches as all other signal contributions need to be known to isolate the diffusion effect. Second, the high scan efficiency comes at the cost of image quality, while the transient nature of the diffusion-sensitized signal makes the acquisition highly vulnerable to brain pulsation and patient motion. Also, in the actual measurement the primary signal evolution, governed by relaxation and diffusion effects, is contaminated by secondary terms from various experimental sources, such as non-Gaussian noise, coherent and incoherent motion, stimulated or spurious echoes due to the interplay of diffusion-encoding and readout gradients – and most of the time a combination of them. These secondary signal contributions are known to cause image artifacts that are spatially correlated. Also, as we use multi-coil imaging and have aliasing due to spiral undersampling, spatial mixing of tissue components is an inevitable consequence of the acquisition scheme. We thus approach the multi-parameter inference task with a CNN architecture to take advantage of all information available, i.e. the temporal and spatial relationships, to characterize the individual signal contributions. Moreover, the deep learning approach benefits from the implicit physical relationships between the scalar and tensorial parameters (Tax et al., 2018; Bernin and Topgaard, 2013) to recover the underlying relaxation and orientational diffusion information.

We have demonstrated that our CNN-based reconstruction framework resolved image corruptions and reliably mitigated pulsation artifacts. Severe head motion however turned out to be the major challenge. As only data from MS patients were affected by severe motion artifacts, we hypothesize that this is because the diffusion-sensitized MRF data from MS patients were acquired at the end of a 1.5 hours scanning session so that patients tended to get restless. The session for healthy controls was comparatively shorter (~40 minutes). Compared to steady-state MRI sequences, MRF relies on transient MR signals. As such, diffusion-weighted MRF schemes are by design more sensitive to motion artifacts than steady-state diffusion-weighted EPI. However, EPI-based DTI suffers from EPI-induced distortions that must be corrected retrospectively. This does not apply to our case: First, we use spiral readouts instead of EPI in the MRF acquisition. Second, due to the significantly shorter timing with monopolar diffusion-encoding gradients, eddy-current induced blurring is reduced compared to approaches based on bipolar gradients. As this is the first study to explore the simultaneous quantification of relaxation and orientational diffusion in an MRF setting, we are confident that we will benefit from the recent advances on how to cope with motion in DW-SSFP – either prospectively or retrospectively.

The proposed MRF framework is based on the diffusion tensor model. Although it is robust and widely accepted, it has the inherent limitation that it fails for crossing fibers.

This shortcoming equally holds for conventional, state-of-the-art DTI methods. Overall, our framework has nevertheless proven to provide relaxation and diffusion tensor maps which agree well with the clinical reference. This might be attributed to the computationally efficient U-Net architecture that has particularly shown convincing performance on small biomedical image datasets. That is, the predictive performance of our model and its ability to resolve even severe motion artifacts could certainly benefit from more training data.

We also anticipate that image artifacts, which mask the fine anisotropic structures, are the main reason why the off-diagonal diffusion tensor elements are not captured as well as diagonal elements. We also believe that a thorough assessment of the individual diffusion encoding directions and their effect on the final diffusion tensor quantification is as important as ameliorating motion artifacts. However, in the proposed MRF scheme diffusion-weightings propagate over multiple repetitions, similar to diffusion SSFP techniques. This results in a mixing of signal pathways which have experienced different histories of diffusion-encoding gradients (strength and direction). With the current dataset, it is thus not possible to retrospectively investigate the effectiveness of the individual diffusion encoding directions that have been applied in full detail. It is thus subject to our current and follow-up work to investigate this in dedicated experiments. We expect that resultant adjustments in the sequence design, specifically in the way we incorporate diffusion-encoding, and techniques such as adaptive spoiling will increase the robustness of the acquisition in first place. Reduced image artifacts, in turn, will enhance the predictive quality of our CNN and allow us to fully regain the characteristic fiber structure in high anisotropy WM regions. We also believe, that proceeding to more advanced deep learning approaches now have a chance to improve on our baseline.

In conclusion, we present a novel MRF-type sequence which simultaneously encodes T1, T2 and orientational diffusion information. We rely on a deep learning-based approach to reconstruct multi-parametric outputs from spatiotemporal MRF data corrupted by artifacts due to spiral undersampling, motion, and the interference of diffusion-encoding and readout gradients. We bypass conventional dictionary matching by learning the intrinsic physical connections between the scalar and tensorial tissue parameters, and thereby propose a scalable MRF application which can be extended to further quantitative contrasts.

## Acknowledgments

Data were acquired as part of a project funded by the MS Society UK. Carolin M. Pirkl is supported by Deutsche Forschungsgemeinschaft (DFG) through TUM International Graduate School of Science and Engineering (IGSSE), GSC 81. Derek K. Jones is supported by a Wellcome Trust Investigator Award (096646/Z/11/Z) and a Wellcome Trust Strategic Award (104943/Z/14/Z). The TITAN Xp GPU used for this research was donated by the NVIDIA Corporation.

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

## Appendix A. Scan parameters

Table 2: Scan parameters. For each of the sequences, the main acquisition parameters are provided.

| | Relaxation and diffusion-sensitized MRF scheme | Clinical reference sequences | | | | |
|---|---|---|---|---|---|---|
| | | DTI | DESPOT1/2 | T1-weighted | PD/T2-weighted | FLAIR (T2-weighted) |
| | | Single-shot diffusion-weighted EPI | SPGR / IR-SPGR / bSSFP | FSPGR | SE | SE / IR |
| Native resolution (mm$^3$) | $1.2 \times 1.2 \times 5.0$ | $1.8 \times 1.8 \times 2.4$ | $1.7 \times 1.7 \times 1.7$ | $1.0 \times 1.0 \times 1.0$ | $0.94 \times 0.94 \times 4.5$ | $0.86 \times 0.86 \times 4.5$ |
| Matrix size | $256 \times 256$ | $96 \times 96 \times 36$ | $128 \times 128 \times 88$ | $256 \times 256 \times 172$ | $256 \times 256$ | $256 \times 256$ |
| Field of view (mm) | 225 | 230 | 220 | 256 | 240 | 220 |
| Slices | 8-12 | 57 | None - 3D | None - 3D | 36 (3mm + 1.5mm gap) | 36 (3mm + 1.5mm gap) |
| TE (ms) | 6 | 94.5 | 2.1 / 2.1 / 1.6 | 3.0 | 9.0 / 80.6 | 122.3 |
| TR (ms) | 22, 50 | 16000 | 4.7 / 4.7 / 3.2 | 7.8 | 3000 | 9502 |
| TI (ms) | 18 | - | - / 450 / - | 450 | - | 2250 |
| Flip angle $\alpha$ (°) | 37, $0 \leq \alpha \leq 49$ ramps | 90 | [3, 4, 5, 6, 7, 8, 9, 13, 18] / [5] / [10.6, 14.1, 18.5, 23.8, 29.1, 35.3, 45, 60] | 20 | 90 | 90 |
| b-values (s/mm$^2$) | - | 1200 | - | - | - | - |
| Gradient amplitude $g_{x,y,z}$ (mT/m) | $-40 \leq g_{x,y,z} \leq 40$ | - | - | - | - | - |
| Gradient duration $\delta$ (ms) | 3 | - | - | - | - | - |
| Spiral interleaves (number) | 34 | - | - | - | - | - |
| Total acquisition time (min) | 4.2-6.4 | 12.5 | 10 | 7.5 | 2 | 3 |

## Appendix B. CNN training progress

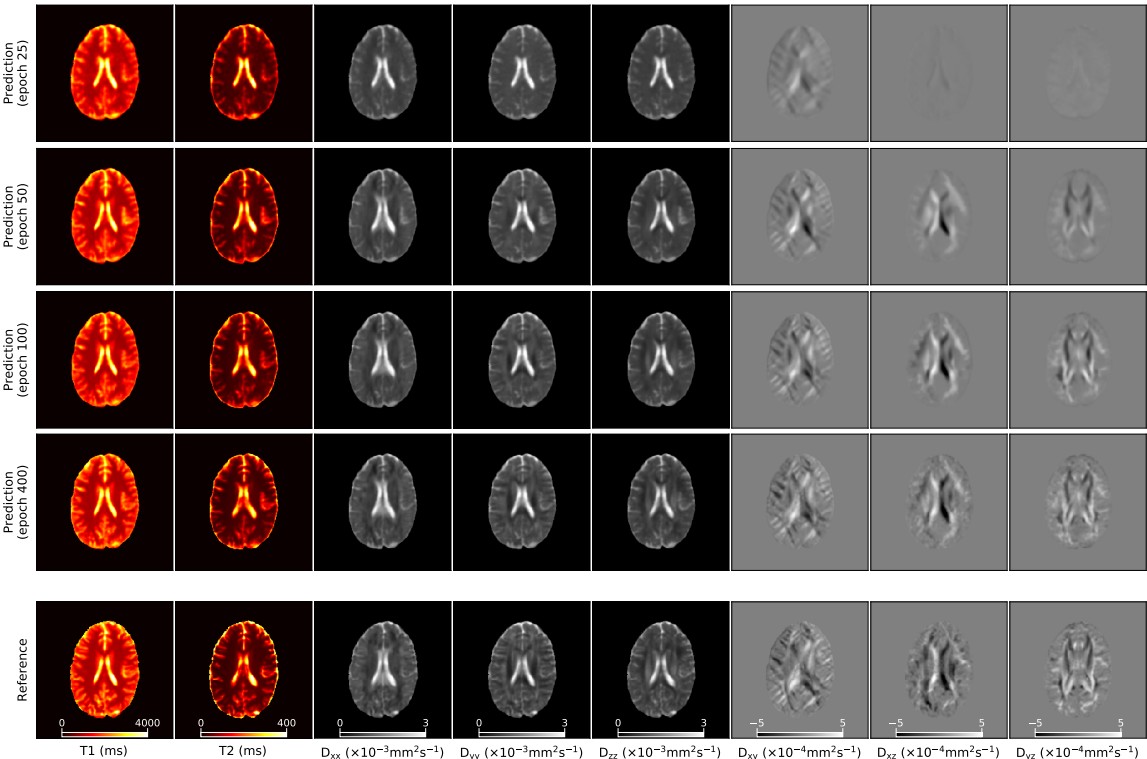

Figure 4: CNN training progress. Predicted relaxation and diffusion tensor maps are shown for increasing number of training epochs (top rows) together with the clinical reference (bottom row). The CNN reconstructs main anatomical structures after a few training epochs while finer structural details of the parameter maps become more pronounced at later stages. With increasing number of epochs, the network learns to gradually recover directional diffusion information in the diffusion tensor maps, eventually revealing the characteristic fiber structures in WM.

## Appendix C. CNN reconstruction in case of severe motion artifacts

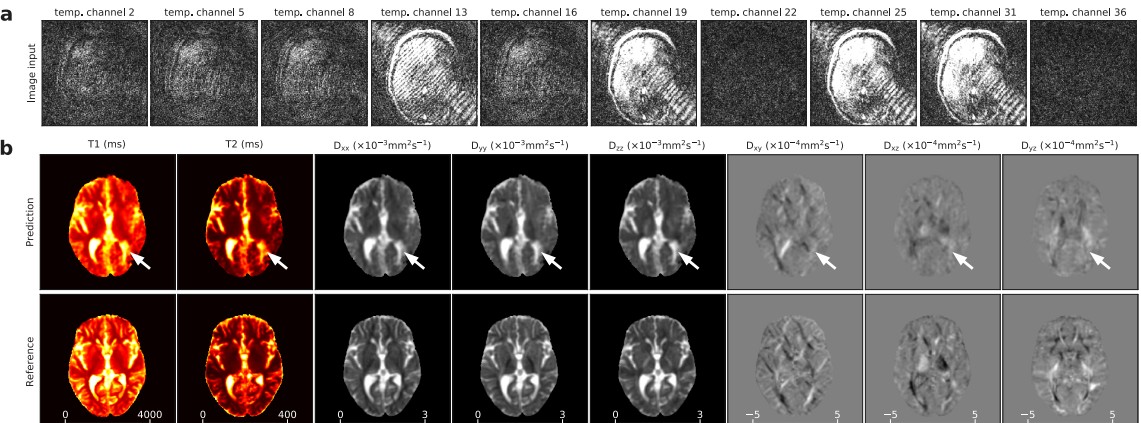

Figure 5: CNN reconstruction for a representative test dataset with severe artifacts. a) MRF images have significant artifacts due to head motion. b) Predicted maps indicate that the CNN was not able to recover relaxation and diffusion information in regions which are severely corrupted (arrow).

