# OpenReview forum: "Deep learning-based parameter mapping for joint relaxation and diffusion tensor MR Fingerprinting"
_MIDL.io/2020/Conference — MIDL 2020_

### Official Review · AnonReviewer1 · 2020-03-02
**The presented work seems interesting for the MRF community but not for MIDL**

**Rating:** 1
**Confidence:** 4

**Summary:**

The authors present a deep learning based reconstruction of quantitative T1 and T2 maps as well as directional diffusion information from diffusion-weighted MRF data. They chose a DL based approach since conventional dictionary based methods do not scale well to larger parameter spaces required for diffusion-weighted MRF. For the reconstruction the authors used a classic U-Net. The approach was evaluated on healthy subjects as well as on MS patient data by comparing the obtained parameter maps to data obtained using the classical approaches for T1/2 MRF and diffusion-weighted MRI.

**Strengths:**

The overall methodology seems sound and the results might be valuable for the MRF community. The approach is much faster than the reference methods. The authors performed validation of healthy as well as patient cases, which is crucial for such learning based image generation approach.

**Weaknesses:**

The deep learning part of the presented work is a straight forward application of an existing method (U-Net) and lacks originality and novelty. This would be ok for an MR focused conference such as ISMRM, but not for a DL focused conference such as MIDL.

The claim "we achieved a comparable reconstruction performance" seems arbitrary. There seem to be significant differences between the results of the proposed method and the chosen reference. It remains unclear how these differences might influence subsequent analyses. Unfortunately there is no phantom based analysis enabling an absolute quantification if the method yields better or worse results then the reference methods.

**Justification Of Rating:**

The paper is methodologically sound but it lacks novelty or originality in the deep learning aspect. Simply applying a U-Net to a problem that is not of very broad interest is not enough to be accepted at a deep learning conference. This work would be suitable for an ISMRM submission since the MR part seems much more interesting.

**Paper Type:**

methodological development

**Special Issue:**

no

---

> ### Author Response · Authors · 2020-03-27
> **Reply to Reviewer 1**
>
> We thank the reviewer and appreciate the thorough review of our work.
>
> 1. Lack of novelty in the deep learning aspect
> We fully agree with the reviewer that the main novelty of our work is not the deep-learning aspect. At the same time we cannot share the reviewer’s central theme of MIDL being a pure deep learning conference, especially since MIDL aims at being a “venue which brings deep learning and medical imaging researchers together for in-depth discussion and exchange of ideas.“ This is exactly, where we see the novelty of our work, namely at the interface of medical imaging and deep learning. The potential that arises when combining the proposed MRF scheme with the deep learning-based multiparametric mapping presents abundant innovation on the topics of “image acquisition, reconstruction and synthesis” and “integration of imaging and clinical data”. We are the first ones to present a joint relaxometry and diffusion tensor MRF framework. We significantly reduce scan time compared to conventional quantitative MR protocols and readily provide multiple quantitative maps in the same image space, making extensive post-processing pipelines of conventional multi-contrast imaging redundant.
> We solve an image reconstruction problem that would be intractable if we only used conventional techniques. That is, with the U-Net architecture we apply an effective model that captures the temporal and spatial relationships of the transient MRF signals. We emphasize that we aimed for “integrat[ing] a novel sequence with a very suitable algorithm” (Reviewer 2) to identify primary and secondary signal contributions to eventually recover the underlying relaxation and orientational diffusion information.
>
> We would also like to bring the reviewer’s attention to prior works at MIDL combining deep learning and MRF and quantitative MRI, respectively, which served as motivation for us. We have now cited them to point out the communities’ interest:
> 1.	Balsiger et al., On the Spatial and Temporal Influence for the Reconstruction of Magnetic Resonance Fingerprinting, MIDL, 2019
> 2.	Golbabaee et al., Spatio-temporal regularization for deep MR Fingerprinting, MIDL, 2019
> 3.	Girardeau et al., Deep Learning for Magnetic Resonance Fingerprinting, MIDL, 2019
> 4.	Chen et al., Deep Fully Convolutional Network for MR Fingerprinting, MIDL, 2019
> 5.	Martini et al., Robust reconstruction of cardiac T1 maps using RNNs, MIDL, 2019
> 6.	Scannell et al., Deep learning-based prediction of kinetic parameters from myocardial perfusion MRI, MIDL, 2019
>
> We sincerely urge the reviewer to reconsider their review by looking at our work from a different perspective of bringing the MRF and deep learning research communities together.
>
> 2. Reconstruction performance and its influence on subsequent analyses
> We fully agree with the reviewer that the results of subsequent analyses are an interesting topic for comparative investigation of our framework and other baseline methods. In fact, we did evaluate our framework based on its direct output as well as secondary measures, namely scalar diffusion metrics and the primary diffusion direction obtained from a subsequent DTI analysis. As seen from Figure 2 and Table 1, our reconstruction performance is comparable with clinical reference methods in terms of relaxation and diffusion tensor quantification. These findings are confirmed when additionally taking the results of the secondary DTI analysis into account, as shown in Figure 3 and Table 1.
> However, we do agree with the reviewer that a more wholistic study with additional subsequent measures would provide more insights. We thank the reviewer for this and plan to include it in follow-up work as this goes beyond our initial proof-of-concept.
>
> 3. Phantom study
> Thank you for pointing this out. It is true that a stand-alone phantom study would allow us to evaluate our framework against a defined ground-truth. However, we don’t think phantom studies are the sole approach for validation, because the measurement conditions in such a very confined, perfect scenario would differ significantly from actual in-vivo scans. Phantom experiments cannot reproduce the non-predictable signal contaminations and artifacts from coherent and incoherent motion, less optimal shim and B1 etc., all of which present major challenges in any multi-shot diffusion acquisition, and thus also diffusion tensor MRF. As such, they are not feasible in our case as phantom experiments‘ significance regarding in-vivo acquisitions and robustness for clinical application is limited.

---

### Official Review · AnonReviewer4 · 2020-03-10
**It's a well written interesting paper but it is unclear how novel it is to a reader unfamiliar with DTI and MRF.**

**Rating:** 2
**Confidence:** 2

**Summary:**

This paper describes a novel MRF pulse sequence that allows for DTI to be acquired alongside T1 and T2.
While standard MRF uses a dictionary matching technique, this paper opts for a U-net for reconstruction of T1, T2, and diffusion tensor images.
The paper shows reasonable concordance between the U-net reconstructed images and the images acquired using standard techniques.

**Strengths:**

The paper is clearly and succinctly written, describing the experiment in adequate detail.
The paper analyzes the performance of the algorithm on reconstructing different maps in different types of tissue.


**Weaknesses:**

The paper claims to be a proof of concept and demonstrates the possibility of such an acquisition scheme but does not provide any comparisons to other methods.
For example, the introduction mentions other diffusion weighted MRF techniques but the paper makes no attempt to compare the technique at hand to them.
The paper would benefit from some kind of baseline comparison or discussion of related techniques.

It is very possible that the U-net could learn a model of a typical image map.
The paper cites the regularizing effect the U-net could be due to the imposition of spatial correlations but it is possible that this could be due to the network's ability to memorize the training data instead.
Some discussion as to why this is not the case would be good.

The paper uses an MS dataset.
RMSEs within the MS lesions are reported but more discussion would be nice.
It seems safe to assume that the MS lesions add heterogeneity to the data.
Demonstration that the network accurately reconstructs these regions would allay concerns about the network's ability to memorize the training data.

**Detailed Comments:**

What are the enlarged portions of the images supposed to show?


**Justification Of Rating:**

The method seems novel and interesting.
However due to the lack of baseline comparisons or an in depth discussion of related techniques, it is unclear the degree to which the method represents an improvement.

**Paper Type:**

methodological development

**Questions To Address In The Rebuttal:**

How does the work proposed compare to previous work?


**Special Issue:**

no

---

> ### Author Response · Authors · 2020-03-27
> **Reply to Reviewer 4**
>
> Thanks a lot for the comments and the possibility to give our views on the review.
>
> 1. How does the proposed work compare to previous work?
> First concepts of MRF acquisition schemes with integrated diffusion-encoding have been presented previously, as we’ve acknowledged in the introduction. All these recent approaches are confined to the estimation of the apparent diffusion coefficient (ADC) together with other MR parameters, while we are the first ones to encode orientational diffusion information to reconstruct the full diffusion tensor - instead of only quantifying mean diffusivity - from a single, rapid MRF-type acquisition. We have now highlighted this substantial difference in the introduction of our manuscript. Consequently, an evaluation of our method against these diffusion-weighted MRF schemes would thus not be conclusive. Therefore, we provide a concise validation of our method against a suitable reference DTI method (single-shot diffusion-weighted EPI).
> Probably we did not emphasize any further differences before, but we do it now: All previously presented diffusion-weighted MRF schemes rely on a dictionary-based approach for parameter inference. This is intractable for our task as we’ve illustrated in our manuscript as well as in the reply to Reviewer 3. Also, with the incorporation of diffusion sensitizing gradients, MR sequences become prone to macroscopic motion and eddy-current induced distortions which particularly complicates the analysis of transient MRF signals. To mitigate coherent and incoherent motion, the previously presented MRF techniques depend on multiple repetitions trying to average these effects out (Jiang et al., 2016) or require cardiac cycle triggering (Jiang et al., 2017). This results in clinical infeasible acquisition times of 2:30min and 60s per slice, respectively. Rieger et al. presented a diffusion-sensitive MRF sequence with a measurement time (28s/slice) that is comparable to our proposed method. However, for dictionary simulation they require B1+ maps to be acquired prior to the MRF acquisition which prolongs the effective scan time.
>
> 2. How does the work compare to quantitative baseline techniques?
> We fully agree with you that a thorough validation against gold standard methods is vital.
> That is why we did evaluate the multiparametric prediction of our framework against clinical reference protocols: As you can see from Figure 2 and Table 1, we explicitly assessed the reconstruction of T1 and T2 relaxation times to the DESPOT1 and DESPOT2 methods, and we compared the predicted diffusion tensor maps and derived diffusion metrics (see Figure 3) with a suitable DTI reference. At the same time, there is already an implicit comparison to existing baseline methods at the training stage as we rely on pairs of MRF image time-series and parametric reference maps.
>
> 3. Generalization ability of the proposed U-Net-based reconstruction
> Thank you for raising this important point. Indeed, we aimed at assessing the generalization ability of our model. Therefore, we intentionally increased the heterogeneity as the training and testing dataset comprises both healthy subjects and MS patients. We performed an in-depth, region-specific analysis where we differentiated between white matter (WM), gray matter (GM), cerebrospinal fluid (CSF) and pathological tissue, namely MS lesions. As seen from Table 1, we reliably recover diffusion and relaxation information, also in regions of diagnostic importance such as MS lesions, indicating generalization capability of our method.
>
> 4. Detailed comments: What are the enlarged portions of the images supposed to show?
> As you can see from Figures 2 and 3, we achieve an image quality in the quantitative maps that is comparable to state-of-the-art methods. The zoomed views demonstrate that we reliably recover fine anatomical details as well as the characteristic WM fiber structure and its primary diffusion direction. We have edited the caption to make this clear.

---

### Official Review · AnonReviewer2 · 2020-03-11
**The paper introduces a novel diffusion MR fingerprinting sequence with a deep learning algorithm for estimating imaging parameters.**

**Rating:** 4
**Confidence:** 5
**Recommendation:** Best Paper Award, Oral

**Summary:**

This paper introduced a diffusion MRI fingerprinting sequence with varying flip angles and diffusion encoding sequences. Then a deep convolutional network was introduced to estimate the parameter maps using imaging data from this novel sequences. Different from previous methods, this paper focused on the estimation of diffusion tensors instead of the mean diffusivity.

**Strengths:**

1) This paper introduces a novel diffusion MRI fingerprinting sequences with varying flip angle and diffusion encoding sequences.
2) The algorithm can simultaneously estimate the relaxation parameters and diffusion tensor.
3) The estimated parameters were validated using experiments.

**Weaknesses:**

1) The diffusion direction shown in Figure 3) seems to be biased.
2) The DTI model is not suitable for crossing fibers, which should be mentioned.
3) The method was trained using healthy subjects. Thus it may be not suitable for analyzing patient data.

**Detailed Comments:**

It is helpful to add some comments on the limitations.

**Justification Of Rating:**

This is very complete and solid work on relaxation and diffusion fingerprinting imaging.
Separate acquisition of diffusion MRI and relaxometry images takes a long scan time. The method introduced in this paper could significantly reduce the scan time. The paper integrate a novel sequence with a very suitable algorithm.  Although it has limitations in practical application, it is still a very novel and interesting method paper.

**Paper Type:**

methodological development

**Questions To Address In The Rebuttal:**

1) Figure 2a temp. channel 13 shows some artifact. Please explain the reason and how that was it handled.
2) Please add quantitative analysis on the bias of principle gradient directions.

**Special Issue:**

yes

---

> ### Author Response · Authors · 2020-03-27
> **Reply to Reviewer 2**
>
> Thank you for the positive comments and the critical suggestions.
>
> 1. Bias analysis of diffusion encoding directions
> We fully agree with the reviewer that a thorough assessment of the individual diffusion encoding directions and their effect on the final diffusion tensor quantification is important. However, in contrast to single-shot EPI DWI, in the MRF acquisition, diffusion-weightings propagate over multiple repetitions in our proposed acquisition, similar to diffusion SSFP techniques. This results in a mixing of signal pathways which have experienced different histories of diffusion-encoding gradients (strength and direction). With the current dataset, it is thus not possible to retrospectively investigate the effectiveness of the individual diffusion encoding directions that have been applied in full detail. It is thus subject to our current and future work, to investigate this in dedicated experiments. Also, we plan to extend the currently available signal models, such as the extended phase graph formalism, by incorporating anisotropic diffusion. Up to now, these models only account for isotropic diffusion which hinders a simulation study for our sequence so far.
>
> 2. Suitability and limitations of the DTI model
> We fully agree with you that although the diffusion tensor model is robust and widely accepted, it has the inherent limitation that it fails for crossing fibers. This shortcoming holds for conventional DTI as well as for the proposed MRF approach. We have now included this in the manuscript in the discussion section.
>
> 3. Generalization ability of deep-learning model
> As an attempt to determine how well our framework generalizes to unseen data and particularly to unseen pathology, such as MS lesions, we ensured that training and testing datasets comprised both healthy subjects and MS patients in the course of the cross-validation experiment. We have now stated this more clearly in the methods section.
>
> 4. Explanation and analysis of artifacts
> The applied encoding scheme is efficient as diffusion-sensitization accumulates over multiple repetitions (like in diffusion-weighted SSFP) and hence echo pathways. However this also leads to secondary signal contributions. We thus see image artifacts that arise from coherent and incoherent motion, aliasing, stimulated or spurious echoes due to the interplay of diffusion-encoding and readout gradients – and most of the time a combination of them. In the specific case of temporal channel 13 in the reconstructed MRF data, we attribute the dominating image artifact to a secondary echo that is caused by the refocusing effect of diffusion-encoding gradients. To ameliorate these secondary signal contributions, we rely on pairs of MRF image-series and corresponding quantitative reference maps, so that our model can learn to identify the primary signal component from all other experimental sources to eventually recover the underlying parameters. To increase the robustness of the acquisition in first place, we will further investigate the way we incorporate diffusion-encoding as well as applying approaches such as adaptive spoiling.
>
> 5. Detailed comments: Limitations of work
> We fully agree with you that a careful discussion of the limitations is vital. We thus clearly state the limitations of our framework and possible solutions in the discussion section. Also, we show an exemplary case of severe motion artifacts, which challenges our method the most, in the appendix.

---

### Official Review · AnonReviewer3 · 2020-03-11
**Intriguing new method allowing magnetic resonance fingerprinting in diffusion tensor imaging**

**Rating:** 4
**Confidence:** 3
**Recommendation:** Poster

**Summary:**

The authors present a fast multi-parameter MRI acquisition based on compressed sensing (MRF). Instead of estimating parameters using dictionary matching, the authors perform parameter regression using a CNN for rich spatio-temporal regularisation. This is the first work (to my knowledge) to successfully measure the full diffusion tensor information with MRF, which the authors claim is made possible by the improved regularisation from their CNN.

The paper's experiments examine data from 20 subjects, 9 healthy and 11 with multiple sclerosis. The authors compare their method with several more standard reference acquisitions (DTI, DESPOT1/2), mostly finding good agreement with their (accelerated) method. This work marks an intriguing step forward in combining advanced MRF-type acquisitions with deep learning based parameter regression, and I believe it would be a valuable contribution to MIDL.

**Strengths:**

Development of an MRF-style acquisition for full DTI information (not just ADC) simultaneous with other parameters. This is among the first such work to be successful in estimating the full DTI information.

Development of a U-Net approach for regularised parameter regression, which builds upon previous successful applications of CNNs in diffusion imaging parameter estimation. The authors claim, convincingly, that this is necessary for successful parameter estimation from their accelerated acquisition.

Clear comparison against suitable reference acquisition methods, across a convincing number of experimental subjects (N=20) with different pathologies (9 healthy subjects, 11 with multiple sclerosis). This gives a higher degree of confidence in the authors' results and their future potential. The examination of MS micro-structural changes also allows for a slightly more in-depth investigation of the method.

**Weaknesses:**

Relatively brief analysis of results. In the authors' defence, they do include several very informative example figures, and their Table 1 is quite rich in information. Nonetheless, it feels as if the paper spends a lot of time on Methods, and not much on Results. Ideally I would like to see more informative comparisons of the different failure modes for different statistics.

Although the authors describe their CNN architecture and pre-processing in reasonable detail, many details would benefit from further justification. The clearest example of this is the "Data pre-processing" paragraph. The authors make several claims about the benefits of their chosen normalisation, but no data are presented to back these up. Ideally there would be an ablation study for some of these details, giving more weight to the authors' claims. In practice, even a discussion of the authors' preliminary experiments while developing the method would be helpful.

In a similar vein, the authors emphasise that dictionary matching becomes intractable as the number of parameters increases. However, they do not provide any estimate for how long this would take in their problem – it would be good to explicitly show this is impractical in their setting, rather than simply claiming it.

**Detailed Comments:**

Higher level comments:

The authors discuss their method's sensitivity to motion, and even provide examples of this in the Appendix. However, they do not make a comparison with the reference methods, which are also sensitive to motion. Could the authors provide more of a comparison for this?

I would be very interested to see an experiment where the network is trained only on healthy subjects and then evaluated on MS subjects, to see how well it can generalise to an unseen pathology. However, this is a lot to to ask for at the review stage, and I will not blame the authors if they choose not to follow up on it.


Lower level comments:

Throughout - The authors repeatedly confuse "principle" vs "principal" when discussing DTI.

Abstract - "i.e. in Multiple Sclerosis lesions", the "i.e." looks kind of messy, I'd be tempted to change it to "for example".

Section 2.2 - "MS lesions were semi-manually segmented [...]" might benefit from more information/citation.



**Justification Of Rating:**

The authors present a meaningful advance in diffusion imaging, at the interface of acquisition and post-processing. They have significant novelty within each of these areas. Their experimental validation is fairly high quality, and their results are intriguing, and promising for future work.

**Paper Type:**

methodological development

**Questions To Address In The Rebuttal:**

I would want the authors to expand on the points discussed under Weaknesses. I would want the authors to provide further analysis of how their parameter estimates differ from the reference methods. I would also want further discussion/justification of the details mentioned above. Ablation studies would be great, but it is understandable if there is not time to run these.

**Special Issue:**

no

---

> ### Author Response · Authors · 2020-03-27
> **Reply to Reviewer 3**
>
> We thank the reviewer for the positive and insightful comments.
>
> 1. Balance between method description and analysis of reconstruction capabilities
> We fully agree with you that a balance between the methodological description and a comprehensive validation is important. We also found a concise description of our method is vital for this proof-of-concept. Albeit the results section provides a comprehensive comparison of our framework against clinical reference methods, we are aware that results section could be more verbose. Given the space limit, we can provide additional figures, e.g. were we compare representative test cases of a healthy subject and an MS patient, in the appendix.
>
> 2. Ablation study and discussion of previous experiments
> In fact, an ablation study would be very interesting and insightful. However, we have decided to present a proof-of-concept demonstrating the potential of our novel diffusion-sensitized MRF sequence when combined with a deep learning-based multivariate regression. That is, we focused on the most relevant aspects of our work regarding both the methodology and its evaluation. Touching upon your question regarding our data processing, we observed that the way we scaled the parametric maps was superior to min-max normalization as it equally reflected the directionality in the off-diagonal elements, with negative and positive value ranges, in the network training. Also, we found that the recovery of fine anisotropic structures improved with the MAE loss compared to the MSE loss. An ablation study along these lines would be interesting. But since we planned to make our joint acquisition and reconstruction the theme of this work, we reserved this study for a follow-up work.
>
> 3. Why a dictionary matching approach is intractable here
> In our joint relaxometry and diffusion tensor MRF scheme, we estimate T1, T2 and the 6 unique elements of the diffusion tensor. To accomplish this with a grid search approach, a dictionary with 156 billion entries had to be generated. Such a simulation is currently intractable with our data servers. Even if feasible, this would lead to a dictionary matching time of 3.5 years per brain using a 128×128×128 image matrix on an Intel Xeon processor E5-2600 v4 (48 cpus), without having considered the time to simulate the dictionary entries.
>
> 4.	Higher level comments
> 4.1 Sensitivity to motion
> In contrast to steady-state MRI sequences, MRF relies on transient MR signals. As such, diffusion-weighted MRF schemes are by design more sensitive to motion artifacts than steady-state EPI-based DWI. However, EPI-based DWI suffers from EPI-induced distortions that must be corrected retrospectively. This does not apply to our case: First, we use spiral readouts instead of EPI in the MRF acquisition. Second, due to the significantly shorter timing with monopolar diffusion-encoding gradients, eddy-current induced blurring is reduced compared to approaches based on bipolar gradients.
> In our study, only data from MS patients were affected by severe motion artifacts. We hypothesize that this is because the diffusion-sensitized MRF data from MS patients were acquired at the end of a 1.5 hours scanning session so that patients tended to get restless. The session for healthy controls was comparatively shorter (~40 minutes).
>
> 4.2 Generalization ability
> We thank the reviewer for this suggestion. We only observed severe motion artifacts for MS patients in the current dataset. To ensure that these artifacts do not bias this kind of experiment, we now plan to include this study in the follow-up work.
>
> 5. Lower level comments
> Thank you for pointing this out. We have changed the manuscript accordingly.

---

### Meta-Review · Area_Chair1 · 2020-04-07
**MetaReview of Paper182 by AreaChair1**

**Rating:** 2

**Metareview:**

While two reviewers are enthusiastic about the paper, I tend to agree with reviewer 1 who points out this contribution does not fit well within the MIDL domain - it's deep learning portion consists of using a straight up U-net. While MRF is  certainly of interest to many in the MIDL community as evidenced by the previous MIDL papers listed by the authors in their rebuttal, each of those papers can be seen to have a more substantial deep learning component (and/or are on the more limited short paper track).

**Paper Type:**

methodological development

**Special Issue:**

no

---

> ### Author Response · Authors · 2020-04-08
> **Reply to Area Chair**
>
> We appreciate the meta-reviewer agreeing on MRF being of interest to many in the MIDL community.
>
> We would disagree that the contribution is in “using a straight up U-net”. Learning is more than defining a (moderately) new DL architecture or a playful analysis of selected parameters and components of a(nother) network.  In fact, critical elements of our learning task are in identifying the domain, the training schedule, the validation, i.e., in identifying the inference task itself.  This is where we see the key innovation and the contribution of this paper.
> We approach the information-encoding in MRF sequence design and the DL-based parameter inference as a coupled task and thereby enable the simultaneous measurement of tissue properties which would be infeasible with conventional MRF schemes.
> We would agree that proceeding to more advanced DL approaches now have a chance to improve on our baseline.
>
> To add additional papers that deal with similar tasks in the wider MIDL community:
> •	Z. Fang et al., “RCA-U-Net: Residual Channel Attention U-Net for Fast Tissue Quantification in Magnetic Resonance Fingerprinting”, MICCAI, 2019.
> •	Z. Fang et al., “Deep Learning for Fast and Spatially Constrained Tissue Quantification From Highly Accelerated Data in Magnetic Resonance Fingerprinting”, IEEE TMI, 2019
> •	P. Virtue et al., “Better than real: Complex-valued neural nets for MRI fingerprinting” IEEE ICIP, 2017.
> •	I. Oksuz et al., “Magnetic Resonance Fingerprinting Using Recurrent Neural Networks”, IEEE ISBI, 2019.
> •	E. Hoppe et al., “RinQ Fingerprinting: Recurrence-informed Quantile Networks for Magnetic Resonance Fingerprinting”, MICCAI, 2019.
> •	F. Balsiger et al., “Magnetic resonance fingerprinting reconstruction via spatiotemporal convolutional neural networks”, MICCAI MLMIR Workshop, 2018.
> •	P.A. Gómez et al., “Learning a spatiotemporal dictionary for magnetic resonance fingerprinting with compressed sensing”, MICCAI Patch-MI Workshop, 2015.

---

### Decision · Program_Chairs · 2020-04-11

**Decision:**

Accept

**Comment:**

Taking all information into account, it was determined that the paper was accepted based on its merit.